# An Automated Fire Code Compliance Checking Jointly Using Building Information Models and Natural Language Processing

Yukang Wang [1], Yue Liu [1,2], Haozhe Cai [1], Jia Wang [1] and Xiaoping Zhou [1,3,*]

[1] Beijing Key Laboratory of Intelligent Processing for Building Big Data, Beijing University of Civil Engineering & Architecture, Beijing 102616, China; 2108550021031@stu.bucea.edu.cn (Y.W.); liuyue18@foton.com.cn (Y.L.); 202007030120@stu.bucea.edu.cn (H.C.); wangjia@bucea.edu.cn (J.W.)
[2] BAIC Foton Automotive Co., Ltd., Beijing 102206, China
[3] Anhui Province Key Laboratory of Intelligent Building and Building Energy Saving, Anhui Jianzhu University, Hefei 230022, China
* Correspondence: zhouxiaoping@bucea.edu.cn

**Abstract:** Fire checking is indispensable for guaranteeing the fire safety of buildings as it reviews the compliance of the building with fire codes and regulations. Automated Compliance Checking (ACC) systems that check building data utilizing Building Information Modeling (BIM) against fire codes have emerged as an active field of research. Substantial efforts have focused on analyzing the properties of the building components. However, the analysis of the spatial geometric relationships of building components has received inadequate attention. The present study proposes a novel ACC system leveraging Natural Language Processing (NLP) techniques to review the spatial geometric relationships of building components in BIM models. First, a framework for a BIM-based ACC system is delineated and decomposes ACC into three constituent subtasks: building model parsing, code knowledge translation, and compliance check result reporting. Second, an approach for structured processing of spatial geometric stipulations in fire codes using NLP is presented to review the geometric relationships between components in building models. Finally, the system's performance is assessed by testing fire code compliance across various building types utilizing BIM models. The empirical findings demonstrate that the system achieves superior recall compared with the manually formulated gold standard, with the ACC system enabling quick, accurate, and comprehensive automated compliance checking.

**Keywords:** fire safety; automated compliance checking (ACC); building information modeling (BIM); natural language processing (NLP); spatial geometry checking




## 1. Introduction

In the Architecture, Engineering, Construction, and Operation (AECO) industry, construction projects are obligated to adhere to fire code provisions [1]. Successful completion of a construction project necessitates checking compliance with all provisions of the fire code. Manual checking of fire code compliance imposes intensive demands on human examiners regarding time, cost, and error-prone processes [2]. In the United States, millions of dollars are spent on manual compliance checks, with each project's checking cycle requiring a minimum duration of several weeks [3]. Automated Compliance Checking (ACC) serves to mitigate the expenditures of time, financial cost, and incidence of inaccuracies associated with manual compliance checks. Investigation into ACC has emerged as an active research domain.

In 1993, the government of Singapore initiated a building code checking system called BP-Expert [4]. An essential problem with this project is the lack of data, because the data source for BP-Expert is a 3D model of the building interior established from 2D drawings. Building Information Modeling (BIM) provides a high-quality data source for building compliance checking [5]. Therefore, BIM offers a solution for ACC to overcome

these limitations. BIM is the digital representation of the physical building, creating a unified digital model by describing the property and geometric information of the building model [6]. In addition, BIM models contain property information and more rich and abstract semantic information that 2D drawings cannot handle. BIM is extensively employed in the design [7], construction [8,9], and management [10,11] phases of buildings.

Until now, numerous researchers have investigated data apposite for developing compliance checks [12–14]. Pertaining to a compliance review, the Korean government started a KBim project in 2013 capitalizing on BIM [12]. The KBim project translates Korean building code into a computer-readable format and institutes a system to support the ACC of BIM models. Amor and Dimyadi [13] reviewed the historical evolution of ACC efforts and elucidated the methods. Additionally, Lee et al. [14] determined the feasibility of ACC in BIM models by surveying six rule-checking applications covering syntactic, semantic, and geometric checks. Despite years of research and efforts, ACC has made advances, though few genuinely functional ACC products grounded in BIM presently exist. The Solibri Model Checker (SMC) [15] is interoperable, utilizing the open Industry Foundation Classes (IFC) standard. Other than SMC, several other systems [16–19] require more advanced technology to implement ACC as they all predicate on the system framework.

Spatial geometric relationships describe the relation and positional configurations between modeled building components within BIM. This encompasses separating, intersecting, adjacent, distances, and relative placements of objects in 3D space. Complex component relationships expand beyond basic intersecting or adjacency to encapsulate intricate hierarchies, nesting, and assemblies of building elements. Existing researchers have made some critical efforts in developing the systems of ACC, but less research has been carried out regarding checking spatial geometry. Except for SMC [15] and FORNAX [18], none of the aforementioned tools deal with building compliance checking efforts in geometry and space. The SMC and FORNAX systems for checking of spatial geometric relationships are merely mentioned, with no details available on their capabilities or efficacy. Effectively parsing and evaluating spatial geometric relationships of building components remains an outstanding challenge. Advanced methods to parse and infer complex component relational data from BIM designs will be instrumental in overcoming current limitations.

This study proposes an ACC system jointly using BIM and Natural Language Processing (NLP) to address inadequate spatial geometry relationship reviews in previous research. The main contributions of this study are concluded as follows:

1.　Proposes a novel framework for a BIM-based ACC system utilizing NLP. The ACC is divided into three core subtasks: building model parsing, code knowledge translation, and compliance check result. Finally, customized visual displays of compliance outcomes are generated to serve users' needs.
2.　Presents a novel method for parsing spatial geometric relationships in BIM models. The method determines the relationships between complex components in the model. The problem of parsing spatial geometric relationships is solved to ensure the full functionality of the ACC system. The BIM model's parsing directly fills the current gap of compliance checks in spatial geometry.
3.　Develops a structured representation of fire codes leveraging NLP. The fire codes are translated into structured logical expressions and form the logic library to enable automated compliance checking. This allows computers to understand and evaluate the fire codes through the logical expressions.

Our system has broad applicability across automated compliance checks for diverse building types across multiple domains. For instance, our framework for the ACC system is generally applicable. The building codes, e.g., fire code, energy code, and design code automated compliance check, can leverage this framework. Our ACC system checks the spatial geometric relationship analysis. The geometric relationship of building components can be checked, regardless of the building type.

To evaluate the ACC system, we validate system performance using BIM model test cases. The experimental results demonstrate the efficacy of the proposed BIM-based

ACC system, which can quickly, comprehensively, and accurately check construction project compliance.

The remainder of this study is organized as follows. Section 2 reviews the related works on the BIM-based ACC system and IFC. Section 3 proposes the method of the BIM-based ACC. Section 4 conducts the experiments presenting the ACC system and analyzes the performance. Section 5 concludes this study.

## 2. Related Work

The ACC of building design has constituted an active research domain internationally for over five decades [13]. The present work focuses on the ACC systems for BIM models. Consequently, the relevant prior literature on ACC systems and BIM data encoded utilizing the IFC standard is reviewed herein.

### 2.1. Study of BIM-Based Compliance Checking Systems

The typical ACC system has two primary data sources: the structured data of building code and the BIM data of the entire building life cycle. The code knowledge of the ACC system is represented by structured data of building code standards. The code knowledge is the basic standard for compliance checking. The BIM data of the entire life cycle of the building can extract the geometry and property information, furnishing the basis for the ACC of space and components of the building. Current ACC systems still have challenges in the automatically translating building code and checking spatial geometry information.

Some previous research has focused on the structured translation of code knowledge. First, a few applications emerged that could process elementary rules, such as mvdXML [19] and IfcDoc [17]. MvdXML and IfcDoc mainly address simple rules with single or limited property relationships or process the model view in BIM. However, additional development work is required for these methods to handle multiple property relationships. Hence, researchers have explored more general expressions of specification clauses. For instance, Pauwels et al. [20] proposed a semantic rule-checking environment that combines IFC with logical semantics for representing specification rules. Hjelseth and Nisbet [21] utilized the Requirement, Applicability, Selection, and Exception (RASE) approach to represent specification rules. Beach et al. [22,23] could convert specification rules to the Semantic Web Rule Language (SWRL) for reasoning by extending the RASE approach to obtain more powerful semantic information. RASE and SWRL improve the flexibility and reusability of specification rules and address complex specification rules to some degree. Nevertheless, ACC still has some limitations in automatically extracting and transforming specification rules, requiring substantial manual transformation processes.

Another essential component of the ACC is the BIM data. IFC, as the primary format for interoperable data transfer, has been implemented in some projects to build internal data models directly through IFC, such as FORNAX [18] and BERA Object Model (BOM) [24]. Other projects use intermediate toolkits to parse IFC data [16]; as the complexity of construction projects increases, so do the size and the complexity of the IFC data. The use of language web technology is another increasingly popular approach that is based on web technologies such as RDF and URI. An Ontology Web Language for IFC (IfcOWL) was developed for representing BIM data using RDF. The RDF graph model shows the IFC data as directed token graphs. The data obtained from IfcOWL can be accessed using the SPARQL query language [25] while supporting semantic reasoning [26].

Another challenging aspect of ACC for BIM-oriented models is reviewing spatial geometry information. In building codes, most of the rules involve the review of geometry and spatial relationships. In addition to the basic validation review [27], achieving the study without treating geometry relationships and other rules is difficult. In most cases, the processing of spatial geometric relations is mainly conducted with dedicated geometry engines, as in FORNAX [18] and BERA [24]. Borrmann and Rank [28] proposed a spatial analysis of building information models with a spatial query language to support the processing of spatial geometric relations. The necessary components that make up an

automatic compliance checking system are available in previous studies. In practice, there is a large gap between the actual implementation of the ACC system and the implementation of the data processing methods before. In particular, achieving a genuinely automatic check in the processing of geometric relations is still impossible.

Extant scholarship has delineated structured knowledge encoding and spatial analysis techniques for ACC systems. However, limitations remain constraining the realization of fully automated compliance checks. Rule encoding approaches thus far demonstrate progress translating simple code requirements, yet remain constrained in handling complex clauses with multiple properties and relationships. Current methods also necessitate extensive manual effort. Regarding BIM data parsing, implemented IFC-based models contend with escalating complexity in construction projects. While ontological representations such as IfcOWL demonstrate promise for semantic queries, realizing advanced deductive reasoning remains an open challenge. Critically, most systems fail to adequately process spatial geometric relationships fundamental to code compliance. Dedicated geometry engines and spatial query languages assist but cannot fully automate spatial analysis. In summary, gaps persist between existing ACC data processing techniques and the capability for genuinely automated compliance checking, especially for spatial geometries. Thus, transcending the limitations of existing ACC system frameworks will necessitate research innovations in the checking of spatial geometric relationships.

### 2.2. Industry Foundation Classes (IFC)

BIM has the capability to integrate and link substantial data associated with the complete building life cycle [29]. BIM serves as a shared knowledge resource for building information, furnishing a reliable basis for entire life cycle decision making [30]. IFC is a digital protocol for data exchange between diverse software, developed and maintained by SMARTA to address interoperability issues in the AECO industry [31]. (SMARTA stands for the SMARTech Alliance, which is the organization that develops and maintains the Industry Foundation Classes (IFC) data model specification. Members of SMARTA include major BIM software vendors such as Autodesk, Bentley, Nemetschek, and Trimble, which all contribute to advancing IFC.) Researchers have developed various IFC-based BIM servers for extracting, translating, and sharing data [32,33]. IFC is defined using the Standard for the Exchange of Product Model Data (STEP) format [34]. STEP employs an object-oriented expression language for succinct and structured object definitions of building models. The IFC standard includes four data model architectures: the resource layer, the core layer, the interoperability layer, and the domain layer. These layers enable the description of information such as geometry, materials, and relationships of the BIM instance model. IFC delineates the building by the essential building elements encompassed, including beams, slabs, columns, walls, and roofs. Additionally, each component can be depicted using different geometric representations such as swept solid, Boundary Representation (B-rep), or body clipping. These varying representations derive from the 3D modeling approach utilized by BIM drawing tools. Figure 1 shows an IFC instance and corresponding entities, relationships, and association mechanisms.

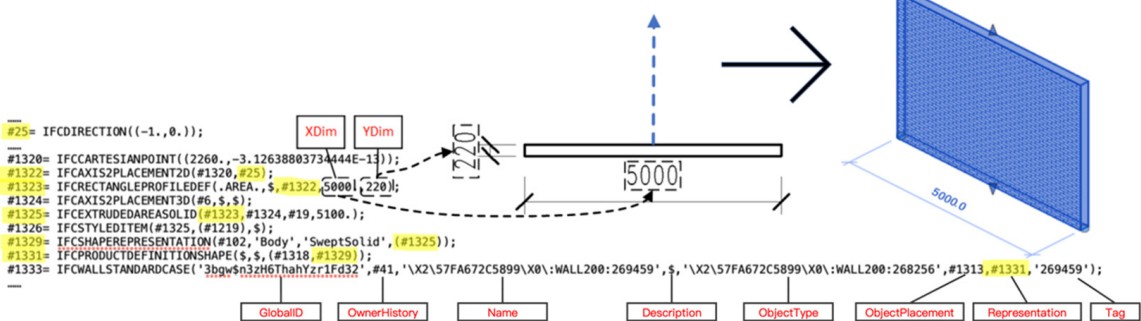

**Figure 1.** Example of an IFC instance.

We can observe that the IFC has a well-structured format and that to query the data efficiently is challenging for such data. A model must be parsed once in sequence and then converted to the target system in the current representation mechanism. In such a scenario, using raw IFC data is very appropriate. However, the data must be queried according to the specification in the ACC system, which requires a more efficient method to retrieve and store the necessary data. In the 1980s, the emergence of Object-Oriented Database Management Systems (OODBMS) presented a solution and some data management systems emerged [35]. But the problem with OODBMS systems is the lack of a standardized query language, which hinders the application of OODBMS. The issue of the structured query language is solved to some extent by Relational Database Management Systems (RDBMS) that represent data as a relationship and transform the data query into a relational question. Thus, the increasing popularity of databases provides more possibilities for processing IFC data.

To enable ACC, checking performance is another aspect to consider when dealing with IFC data. Due to the large number of entities in the IFC (653 entities and 327 types in IFC2×3767 entities and 391 types in IFC4), a one-to-one mapping of IFC entities to a relational database is performed [36]. Retrieval and traditional relational databases are stored in a table structure. Even simple queries will link a large number of tables, which increases as the variety of data increases. The maintenance cost of the database keeps expanding, which will become a bottleneck in the review performance. Graph databases solve this problem by storing data as relational and entity "first-class citizens" of the database. In the spatial geometry of the underlying data, such as determining the distance between two components, this information is usually not directly available in the IFC. One way is to obtain it through automatic calculation or manual addition in the original BIM design software. This not only adds a burden to the modeler but also the limited capability of the BIM design software; this method is generally impractical and difficult to apply. The alternative approach is to develop or formulate specialized spatial and geometric calculation interfaces that can address the limitations of IFC data regarding information acquisition for spatial geometric relationships.

The IFC provides a well-structured format for representing and exchanging BIM data. Efficiently querying on IFC models poses salient challenges. Early systems relied on sequential parsing of IFC files for translation. However, ACC necessitates targeted retrieval based on building codes. Storing IFC in graph databases circumvents the problem of sequential parsing; nonetheless, deriving requisite spatial geometric relationships remains a prominent obstacle. IFC intrinsically lacks connectivity between modeled components, compelling manual augmentation in BIM tools or custom spatial processing modules. This not only overburdens design modelers but also contends with inherent software limitations. In conclusion, the problem that spatial geometric relationship cannot be automatically extracted constrains the development of a robust high-performance ACC system. This underscores the necessity of continued research and development of ACC systems for automated compliance checking of spatial geometry relationships.

## 3. ACC Method for BIM Models

This section proposes a new spatial geometry checking method to address the challenges of handling spatial relationships in an ACC. We present an ACC approach for BIM models, including (1) a framework for automated BIM compliance review, (2) the parsing of BIM model data, (3) code knowledge translation, and (4) compliance check results. In the following, we show the details of the proposed approach with some examples.

### 3.1. Framework

The ultimate goal of ACC is to attain a fully automated system [27]. Experts can then devote more energy to enhancing building performance. A typical ACC system must analyze and process two key data sources: building models and code knowledge. Although

specific systems [37] may have different implementations, a generalization indicates that constructing a rule review system generally encompasses three subtasks.

- Building model parsing: Building models are digital representations of design data and entities subject to a compliance check. Parsing the building model is critical for more comprehensive compliance checks. The parsed data should include the property information of the building model and other information required for the review as much as possible.
- Code knowledge translation: Fire codes provide the foundation for the ACC system. Human-readable code provisions must be translated into computer-processable rules. This translation process can be achieved by human interpretation of the code provisions and converting them into structured provisions or by using methods such as NLP [3,38].
- Compliance check results: By matching the code knowledge with the parsed building information data, the compliance check steps are executed in the system to obtain the final inspection results. The related inspection results are presented as inspection reports, including violated code provisions and associated review objects.

In this section, the framework for the ACC system using BIM is proposed based on the characteristics of the rule review system. Figure 2 shows the framework process and the subtasks in the ACC system process. The building model parsing is either manual input of information data or direct derivation. Information should be parsed or added automatically as much as possible to reduce the risk of manually inputting incorrect data. Code knowledge translation is represented using various semantic models, such as XML formats (LegalRuleML [39], BPMN [40]), meta-language formats (KBimCode [12]), and standard query or rule languages (SWRL, SPARQL [25], Cypher). Compliance check results generally contain pass, fail, and unclear. Since the framework is generic, the subtasks can be supplemented appropriately in future studies.

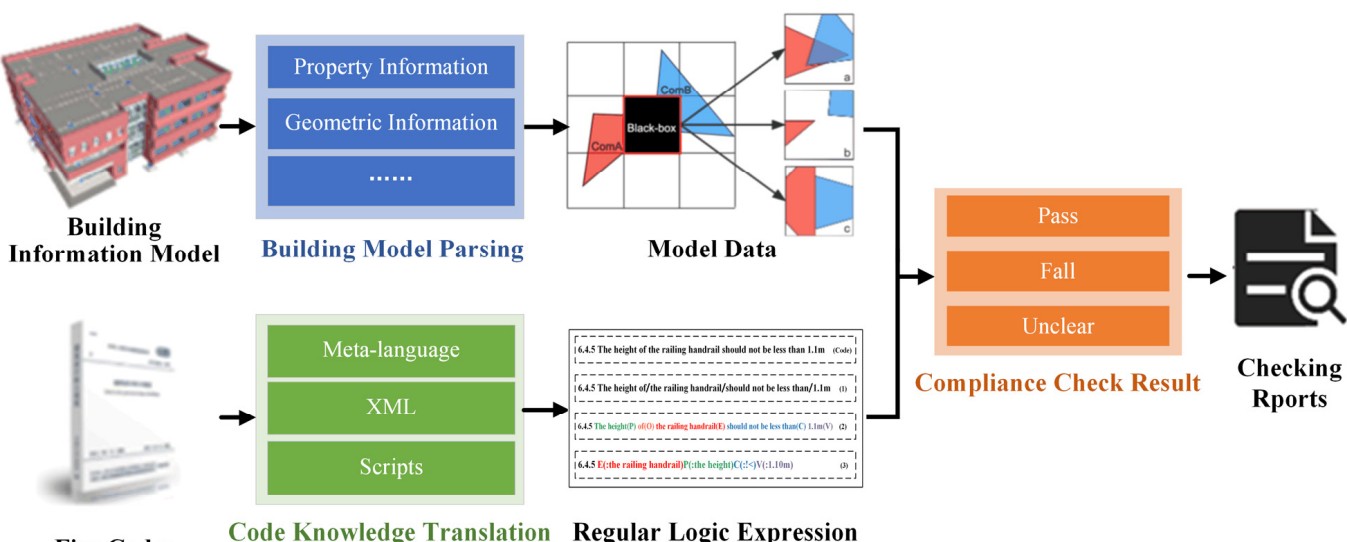

**Figure 2.** Common framework for ACC system using BIM.

### 3.2. BIM Model Paring

After years of research and efforts, the ACC system for BIM models has been gradually improved. Since BIM data are the primary data source for the ACC, processing BIM data is particularly important. The SMC system provides a method for compliance checks of open IFC, serving as a pioneer of ACC systems [25]. BuildingSMART [17], led by the International Chamber of Commerce (ICC), enhances the platform's scalability based on the SMC system. At this stage, parsing property information from BIM models has become relatively mature.

Parsing the geometric relationships between components in a BIM model presents challenges in processing BIM data. Fire codes have many constraints on component geometry and spatial position, in particular, the judgment of the relative position between two components. Comprehensive ACC is only achievable by analyzing geometric relationships in BIM in addition to primary property information. In this section, based on the characteristics of BIM model data, we propose a geometric relationship resolution method for BIM models.

Figure 3 shows an example of the parsing method. The complex 3D component entities cannot determine their spatial location based on property information. Thus, component models typically require meshing, which aims to translate irregular component entities into more regular geometric shapes. A triangular mesh is a volume enclosed by triangles, possessing adaptability, rapid generation, and simple construction advantages. In two-dimensional graphics, complex polygons can be translated to generate triangles. Therefore, this paper first triangulates the building components to represent the geometric shape of an irregular model. The minimum triangular cells constituting the triangular grid are defined as follows:

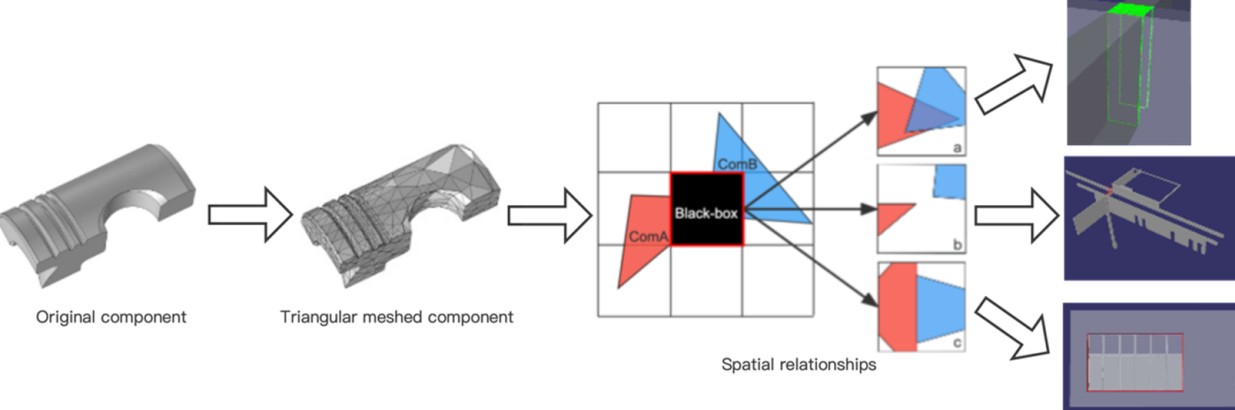

**Figure 3.** Example of geometric relationship resolution.

**Definition 1.** *Triangular Unit (TU) is a triangular group Tri = (point, line, area), where the point is the vertex, the line is the edge, and the area is the face enclosed of the triangular unit, respectively.*

**Definition 2.** *The spatial locations between components are intersecting, adjacent, and separating. Intersecting components represent that the components are overlapping in spatial. Adjacent components indicate that the components are in contact but do not overlap in spatial. Separating components mean that the components do not overlap nor are they in contact.*

The finite number of TUs can dissect the arbitrary building components. The triangular mesh model of the building components is represented in the form of equations as

$$Com = \bigcup_i^n Tri_i, n > 0, \tag{1}$$

where *Com* denotes the building component and *Tri* is the TUs defined in Definition 1. By triangulating the components, the geometric relations of building components are translated into the extraction of TU sets, simplifying the computational difficulty of dealing with irregular components. Figure 4 demonstrates that the TUs are regularly dissected into finite meshes by the grid space. By selecting different spatial mesh scales, the accuracy can be adjusted by extracting the geometric relationships of the components.

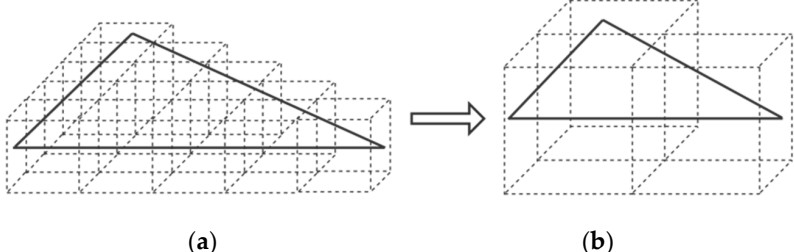

**Figure 4.** Example of TUs for space meshing. (**a**) Scale = 10. (**b**) Scale = 30.

Since the mesh is in 3D space, we project the component triangular mesh to the X0Y plane to more intuitively judge the relationship between the two components. We can intuitively understand the definition of the three spatial positions through the relationship of the components in the 2D plane. Figure 5 shows that the binary coordinate sequence obtained under the 2D coordinate system is coded as $(i_x, i_y)$ and the number of lines in the 3D space is used to form a ternary code of $(i_x, i_y, i_z)$ in the triaxial direction.

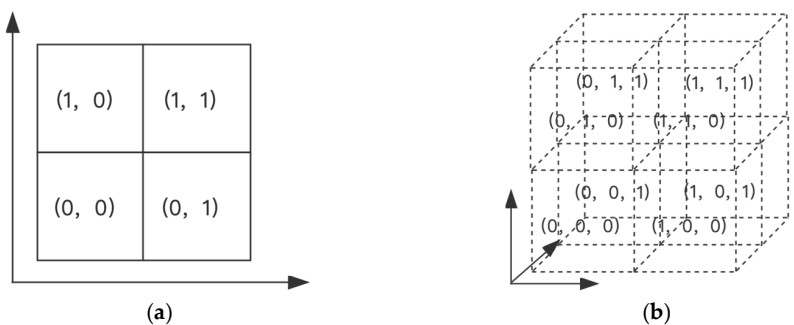

**Figure 5.** Meshing code: (**a**) 2D coordinate system; (**b**) 3D coordinate system.

We specify the spatial locations between components as intersecting, adjacent, and separating as Definition 2. Figure 6 shows the three spatial location relationships. Figure 6a shows that the spatial coding set of component A is ComA = {(0, 0), (1, 0), (1, 1)} and of component B is ComB = {(1, 4), (1, 5), (2, 4)}. Since there is no common coding area, member ComA is separate from ComB and the minimum distance of the components can be calculated. Figure 6b shows that the spatial coding set of component B is ComB = {(1, 1), (1, 2), (2, 1)}. Since there is a standard area (1, 1), the spatial location of the two components intersects. Figure 6c presents the spatial code set of component B as ComB = {(0, 1), (1, 1), (0, 2), (1, 2)}. Since there is a common edge AB, the two members are adjacent.

In studying the distances of the building components, we first select the small component, defined as component A, and then the other one, defined as component B. Then, point clouding is performed for the smaller area of TU. The distance between the two components is translated according to the idea of "component to component → TU to TU → point cloud to TU." We introduce the separation method of the triangular plane, where the three sides of a triangle are extended to form an extension line. The sides of the triangle and its extension lines divide the plane area into seven plane areas.

Figure 7 shows the three vertices $T_1$, $T_2$, and $T_3$ of $TU_1$; their connecting lines and extension lines divide the plane $\pi$ into seven planes. The $P_1$ is the point in the point cloud of $TU_2$. To compute the shortest distance $d_1$ from $P_1$ to $TU_1$, $P_1$ is first projected into triangular plane 1 to obtain projection point $P'_1$ and projection distance $s_1$. Then, the vertical line of the triangle edge $T_1T_3$ obtains point $D_1$ and sheer distance $l_1$. Finally, we compute the shortest distance, termed $d_1$. Therefore, we need to find the distance for each point of each TU in component B. The distance from the point to component A is

$$d_i = \sqrt{s_i{}^2 + l_i{}^2}. \tag{2}$$

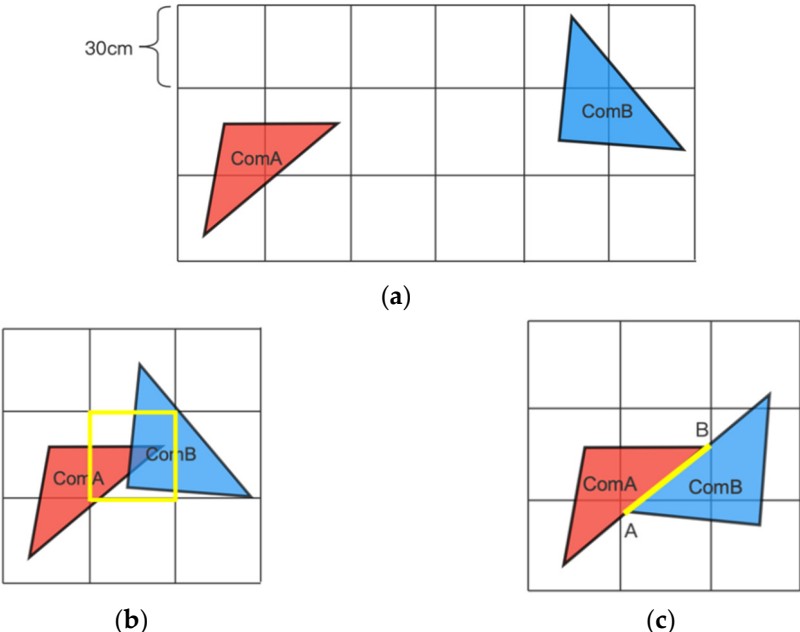

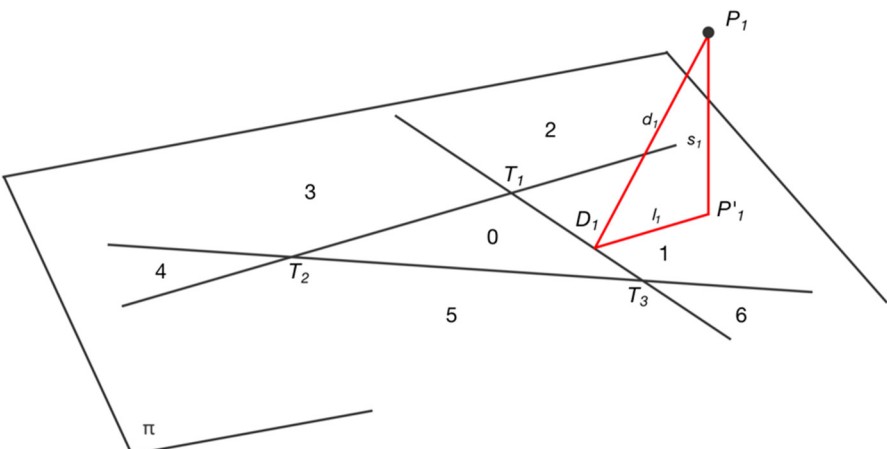

**Figure 6.** Example of spatial relationship. (**a**) Separating. (**b**) Intersecting. (**c**) Adjacent.

**Figure 7.** Example of point-to-plane distance.

Considering that the intercomponent distance is used to support the ACC system using BIM, the distance between the two components should be the shortest distance. Taking the minimum value of the distance from each point to the building component sought, the shortest distance between component A and B is defined as

$$D = mind_i, i \in (1, 2, 3 \cdots n),$$ (3)

where $D$ presents the shortest distance and $n$ is the number of TU.

### 3.3. Code Knowledge Translation

The fire codes are represented by a natural language approach, which is humans' most dominant representation of code understanding. We leverage NLP technology to construct structured logical representation of the fire codes in order to enable computers to comprehend the knowledge information expressed in fire codes. This structured logical representation of the codes is necessitated to buttress compliance checks in the ACC system.

We analyze over 10,000 code clauses in the GB50016-2014 Code for Fire Protection in Building Design. The formalizable provisions are extracted and the meaningless clauses

that require human involvement in judgment are removed. By analyzing more than 4000 formally translatable sentences, the steps of code knowledge translation are divided into (1) word separation processing, (2) entity identification, and (3) formation of regular logical expressions.

The word separation processing of code clauses is a prerequisite for translating code knowledge. A complete code clause is usually complex. First, we must break up the long complex sentences into short ones. Then, the short sentences are split into one word after another. Figure 8 shows the proposed approach's process, further demonstrated by the example of clause 6.4.5. The code clause 6.4.5 states that the outdoor evacuation stairs should comply with the following provisions: (1) The height of the railing handrail should not be less than 1.10 m and the net width of the stairs should not be less than 0.90 m. (2) The inclination angle should not be greater than 45°. The code provisions will be ",", ".", ":", ";" and other punctuation marks for division. The result of the splitting is

$$R = \{r_1, r_2, r_3 \cdots r_n\}, \tag{4}$$

where $r_n$ is the $n$th phrase obtained after segmentation. The corresponding specification 6.4.5 can be expressed as $R = \{r_1$: "Outdoor evacuation stairs should adhere to the following stipulations", $r_2$: "The height of the railing handrail should not be less than 1.10 m", $r_3$: "The net width of the stairs should not be less than 0.90 m", $r_4$: "The inclination angle should not be greater than 45°"}. To better summarize the structured law of the fire code, each phrase can be formally expressed in the form of "[word 1] + [word 2] + [word 3] $\cdots$ + [word n]". Since the code terms belong to domain-wide knowledge, a firefighting domain dictionary needs to be added to the partitioning process. After the final word separation process, the following results were obtained as

$$r = \{v_1, v_2, v_3 \cdots v_i\}, \tag{5}$$

where $v_i$ denotes the $i$th word obtained through the word division.

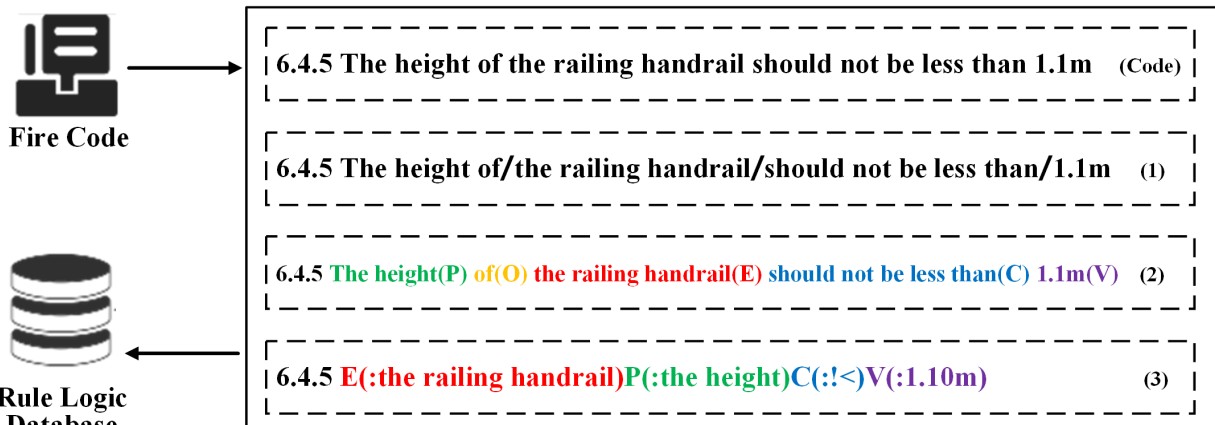

**Figure 8.** Example of the code knowledge translation process. Red indicates component elements, green indicates property information, blue indicates comparative words, and purple indicates specified values.

To entity identification, we combine the rules of defining the component entities in the IFC standard. The code clauses are divided into five categories of entities: component element, property information, space information, comparative word, and specified value, as shown in Table 1. The component element refers to the subject of the check in the code clause, which is expressed as an IFC component entity corresponding to the fire code checks. Property information refers to the properties of the subject of the fire protection provisions, such as length, width, height, etc. Intersection, adjacency, and separation are the three main categories of space information. Comparative words are the words to judge

the relationship between them. Specified value refers to the values in the fire code clauses, generally expressed as a numerical value such as 1.1 m.

**Table 1.** Entities of fire egress codes.

| Classification | Definition | Description |
|---|---|---|
| Component Element | E | Corresponding to the component entities |
| Property Information | P | Properties of the entity |
| Space Information | S | Intersection, adjacency, and separation |
| Comparative Word | C | =; !=; >; !>; <; !< |
| Specified Value | V | Numerical value of code |

The next step is the structured representation (rule logic expressions) of the code provisions. Table 2 shows that we divide the rule logic expressions into three categories: (1) property judgment class, (2) geometric space judgment class, and (3) compound judgment class.

**Table 2.** Regular logic expression.

| Classification | | | Regular Logic Expression | Rule Description |
|---|---|---|---|---|
| Property Judgment Class | Single | | E(: a)P(: b)C(: c)V(: d) | Property b of component a should satisfy the specified value d of condition c. |
| | Multiple | | E(: a)P(: b)C1(: c)V1(: d) and/or C2(: e)V2(: f) | Property b of component a should satisfy the specified value d of condition c and/or should fulfill the selected value f of condition e. |
| Geometric Space Judgment Class | Distance Constraint | Single | E1(: a)E2(: b)S(: distance) C(: c)V(: d) | Component a and component b are separated from each other and the distance should meet the specified value c of condition d. |
| | | Multiple | E1(: a)E2(: b)S(: distance) C1(: c)V1(: d) and/or C2(: e) V2(: f) | Component a and component b are separated, the distance should meet the specified value of c and condition d and/or should meet the fixed value e of condition f. |
| | Inclusion Relationships | | E1(: a)S(: contain)E2(: b) | Component a contains component b. |
| | Position Constraints | | E1(: a)S(: adjacent. up/down/left/right)E2(: b) | Component A and component b are adjacent to each other. |
| Compound Judgment Class | | | E1(: a)S(: adjacent/contain/ separation)E2(: b) and/or E(: a/b)P(: c)C(: d)V(: e) | Component a is adjacent to/contains/is close to component b and/or the property c of component a/b should satisfy the specified value e of condition d |

(1)　Property judgment class. We take the code clause "The height of the railing handrail should not be less than 1.10 m" as an example. The "railing handrail" is the component entity E, "height" is the property information P of this component entity, "shall not be less than" is the comparative word C, and "1.10 m" is the specified value V. Therefore, the rule logic expression is "E(: railing handrail)P(: height)C(: !<)V(: 1.10 m)".

When two specified values are in the code in a special case in the property judgment class, the correct structured representation of the code is needed with the help of logical correlations "and" and "or". We take the code clause "The step of the ladder section shall not exceed 18 steps and shall not be less than 3 steps" as an example. Two constraints are on the "steps of the ladder section", which should meet both "shall not exceed 18 steps" and "shall not be less than 3 steps". Therefore, the rule logic expression is expressed as "E(: ladder section) P(: step) C1(: <)V1(: 18 steps) and C2(: >)V2(: 3 steps)".

(2) Geometric space judgment class. As defined in Section 3.2, spaces consist of intersection, adjacency, and separation. In regular logical expressions, geometric spaces are defined as three classes of distance constraints, inclusion relation classes, and position constraints. The rule logic expressions for positional constraints and inclusion relations are relatively straightforward, e.g., the code clause states "smoke extraction equipment shall be provided in the courtyard" can be expressed as "E1(: courtyard)S(: contain)E2(: smoke extraction equipment)".

For the distance constraint class, comparison words and specified values are appended. For example, for the code clause "The distance between two safety exits shall not be less than 5 m", the two components being compared are "two security exits". The "distance" denotes the spatial information, while "Shall not be less than" and "5 m" are comparison words and specified values. Hence, the rule logic expression is "E1(: security exit)E2(: security exit)S(: distance)C(: !<)V(: 5 m)". Complex multi-distance constraints also require a structured representation using logical correlations such as "and" and "or."

(3) Compound judgment class. The compound judgment class refers to code provisions encompassing both property and geometric space judgment. For example, the code clause states "The warehouse should be set up with 2 safe exits". This clause can be divided into geometric space judgment class clauses and property judgment class clauses. "The warehouse contains a safe exit" and "the number of safe exits is 2". In this case, the rule logic expression would be "E1(: warehouse)S(: contains)E2(: safe exit) and E(: safe exit)P(: number)C(: =)V1(: 2)".

*3.4. Compliance Check Results:*

Fire codes impose many constraints on the property information and geometric relationships of the building components. We match the BIM model parsing information in Section 3.2 with the rule logic expression of fire codes in Section 3.3 and derive the compliance check results. The compliance check results of fire codes generally contain pass, fail, and unclear, denoting the ACC system result.

The check result "pass" signifies that the parsed information component model information satisfies the constraints of the fire codes. For instance, the rule logic expression of "the distance between two safety exits shall not be less than 5 m" is expressed as "E1(: security exit)E2(: security exit)S(: distance)C(: !<)V(: 5 m)". The geometric relationship parsed from the BIM model is that the distance between ComA and ComB is 8 m, which satisfies the constraints of the fire codes. The "fail" result denotes that the property information and geometric relationships of building components contravene the rule logic expressions of the fire codes. For example, the distance between ComA and ComB is 4 m, registering the rule logic expression that violates the fire codes constraints. The "unclear" result signifies no applicable fire codes constraining the corresponding components' property information and geometric relationship.

## 4. Experiment

*4.1. Subsection Experiment Setup*

4.1.1. System Implementation

Our ACC system using BIM is implemented in a proof-of-concept prototype. We opt for a B/S (Browser/Server) grid structure to design and develop the BIM automatic compliance check system.

We test the ACC system in an environment with an Intel Core i7 CPU, 16GB RAM, GTX1660ti discrete graphics card and Windows 10 64-bit operating system to validate our ACC system. The BIM model parsing algorithm is implemented in Java. The code knowledge translation algorithm is realized in Python. Figure 9 shows the user interface of the BIM-based ACC system. The interface consists of two areas: (1) model display and (2) check results, which visualize the outcomes of the ACC in the BIM model.

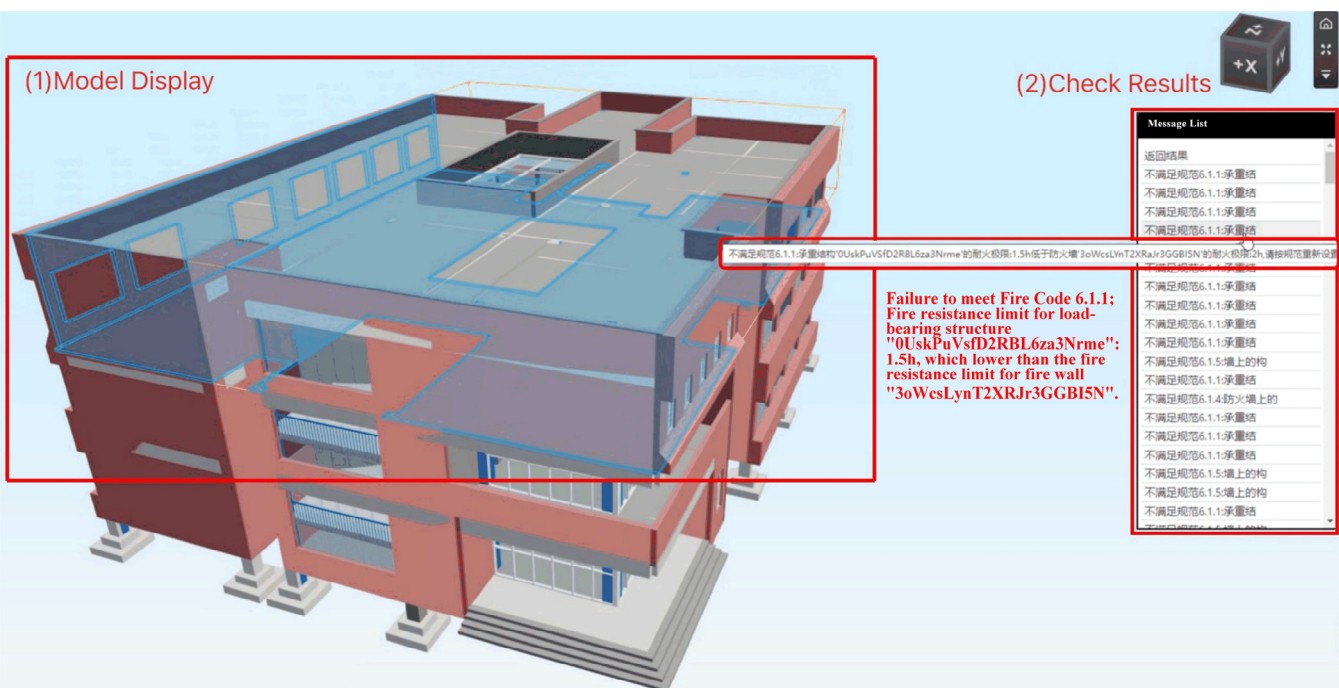

**Figure 9.** Example of ACC system using BIM.

4.1.2. System Test

We select BIM models as the system test data in the ".ifc" format and we test BIM models for 10 different building types. Table 3 shows the specific project information that covers all building types.

**Table 3.** Object information of experiment case.

| Building Type | Instance Number | Item Name | Building Height | Item Size | IFC Data Size |
|---|---|---|---|---|---|
| Public Building | 1 | Academic Building | 13.5 m | 12.0 MB | 4.1 MB |
| | 2 | Supermarket | 5.4 m | 8.3 MB | 2.6 MB |
| | 3 | Office Building | 11.4 m | 13.4 MB | 38.9 MB |
| | 4 | Hotel | 16.9 m | 5.37 MB | 11.8 MB |
| | 5 | Station | 4.2 m | 15.0 MB | 17.2 MB |
| Residential Building | 6 | Villa A | 17.1 m | 8.6 MB | 8.4 MB |
| | 7 | Villa B | 11.5 m | 18.2 MB | 9.6 MB |
| | 8 | Villa C | 10.4 m | 12.9 MB | 8.8 MB |
| | 9 | Apartment | 15.8 m | 12.8 MB | 5.3 MB |
| Industrial Building | 10 | Gas Station | 7.2 m | 20.4 MB | 42.7 MB |

In the experimental validation, the code provisions in Chapter 6 of the GB50016-2014 Building Design Fire Code are randomly selected. We choose to check the BIM model for both properties and spatial geometric relationships. As an example, we take fire code chapter 6.1 of the fire codes, the firewall code provisions, with Table 4 showing the contents to be checked.

**Table 4.** Contents to be checked of experiment case.

| Review | IFC Entity Type | Review Type | Review Content |
|---|---|---|---|
| Beams | IfcBeam | property<br>space | Fire Resistance Limit<br>Location; Distance |
| Floor slab | IfcSlab | property<br>space | Fire Resistance Limit<br>Location |
| Column | IfcColumn | property<br>space | Fire Resistance Limit<br>Location; Distance |
| Wall | IfcWall | property<br>space | Width; Height<br>Location; Distance |
| Firewall | IfcWall | property<br><br>space | Fire Resistance Limit; Fire<br>Rating<br>Location; Distance |
| Window | IfcWindow/<br>IfcWallStandardCase | property<br>space | Width; Height<br>Location; Distance |
| Fire Window | IfcWindow/<br>IfcWallStandardCase | property<br><br>space | Fire Resistance Limit; Fire<br>Rating<br>Location; Distance |
| Door | IfcDoor | property<br>space | Width; Height<br>Location; Distance |
| Fire Door | IfcDoor | property<br><br>space | Fire Resistance Limit; Fire<br>Rating<br>Location; Distance |

4.1.3. Evaluation Metrics

The ACC system evaluates the proposed method regarding accuracy, recall, and F-value. We invite industry experts to perform a manual review of the BIM model and the review results are used as the gold standard. Table 5 shows the test results obtained from this experiment. In the table, Gold Standard Total (G.T.) indicates the number of components that do not meet the code requirements in the gold standard. System Total (S.T.) shows the number of components returned by the computer that do not meet the criteria. True Positives (T.P.) indicate the number of components that are correctly detected in S.T.

**Table 5.** Fire checking examination results.

| Building Type | Instance Number | Item Name | Number of Components | G.T. | S.T. | T.P. |
|---|---|---|---|---|---|---|
| Public Building | 1 | Academic Building | 1086 | 220 | 233 | 213 |
| | 2 | Supermarket | 150 | 106 | 114 | 103 |
| | 3 | Office Building | 429 | 417 | 421 | 409 |
| | 4 | Hotel | 547 | 134 | 149 | 128 |
| | 5 | Station | 22 | 17 | 22 | 17 |
| Residential Building | 6 | Villa A | 2105 | 343 | 389 | 334 |
| | 7 | Villa B | 246 | 157 | 173 | 154 |
| | 8 | Villa C | 839 | 182 | 199 | 166 |
| | 9 | Apartment | 886 | 165 | 188 | 161 |
| Industrial Building | 10 | Gas Station | 628 | 128 | 136 | 126 |

We set precision, recall, and F-value as evaluation metrics to evaluate the experimental results. We term the precision as *P*, defined as

$$P = \frac{\text{Number of correct components in the test results}}{\text{Number of all components in the test results}} \times 100\%, \tag{6}$$

where *P* represents the percentage of building components with a "pass" check result out of the total number of components. We term recall as *R* and *R* computed as

$$R = \frac{\text{Number of correct components in the test results}}{\text{Number of all components in the gold standard}} \times 100\%, \tag{7}$$

where *R* indicates the percentage of building components that are correctly identified as passing compliance out of all components that should pass per the gold standard. The high recall rate signifies the compliance checking result of the ACC system more closely to the manual review by the industry expert. Thus, the recall rate is more important than the precision, which only identifies passing building components. We term the F-value as *F*, which presents the significance of the check result to the gold standard. *F* is defined as

$$F = \frac{2 \times P \times R}{P + R} \times 100\%. \tag{8}$$

*4.2. Subsection Experimental Conclusion*

Ten iterative evaluations are conducted utilizing BIM models as test data, with mean values adopted as the aggregated experimental results. Table 6 presents the assessment outcomes derived from the defined evaluation metrics.

**Table 6.** Test results of evaluation metrics.

| Instance Number | Item Name | P (%) | R (%) | F (%) |
|---|---|---|---|---|
| 1 | Academic Building | 91.41 | 96.81 | 94.03 |
| 2 | Supermarket | 90.35 | 97.16 | 93.63 |
| 3 | Office Building | 97.14 | 98.08 | 97.61 |
| 4 | Hotel | 85.90 | 95.52 | 90.45 |
| 5 | Station | 77.27 | 100 | 87.17 |
| 6 | Villa A | 85.86 | 97.37 | 91.25 |
| 7 | Villa B | 89.01 | 98.08 | 93.33 |
| 8 | Villa C | 83.41 | 91.20 | 87.13 |
| 9 | Apartment | 85.63 | 97.57 | 91.21 |
| 10 | Gas Station | 92.64 | 98.43 | 95.45 |

In Table 6, the precision, recall, and F-value of the test results for the academic building item are 91.41%, 96.81%, and 94.03%. The results of the combined 10 test items show the recall rate can reach (100%) for BIM models with a small number of components and the recall rate is mostly higher than 95% for complex BIM models. The average recall rate (96.81%) of the test results of this project is higher than the average precision rate (91.41%), indicating that our ACC system has better adaptability. The full experimental results demonstrate that our ACC system has a high recall rate.

SMC [15] and FORNAX [18] (mentioned in the related work) only analyze the spatial geometric relationships of building components. In comparison to SMC and FORNAX, our ACC system can automate checking the complex spatial relationships of building components in BIM models. The check results correctly display the component numbers and code clauses conforming to and violating the code requirements. A modification suggestion is also provided for each non-conforming component entity. Our ACC system provides more generalized capabilities by integrating multiple approach strengths while overcoming restrictions of ACC systems jointly using BIM and NLP.

## 5. Conclusions

Implementing the ACC system has always been a crucial task for ensuring the safe operation of construction projects. In this paper, a general framework for ACC is proposed. Based on this framework, a BIM-based ACC system utilizing NLP is proposed. In this ACC system, the spatial geometric relationship resolution method guarantees the comprehensive BIM data processing. More specifically, the building model parsing and code knowledge translation method resolves the automatic check of property information and geometric relationship information of BIM data. Additionally, formulating rule logic expressions provides the code processing capability of the ACC system.

Testing the BIM model demonstrates that our ACC system is promising, with high precision, recall, and F-value performance. Our ACC system provides a generalized solution to enhance and accelerate automated compliance checks of building codes throughout the building life cycle. The spatial geometric resolution method proposed in this paper assists in addressing issues in collision detection, presenting a potential future research direction.

However, our ACC system has some limitations. The logical expressions need further expansion to encapsulate the vast number and broad scope of fire codes involved, especially for compliance checking at the semantic level. We have only initially demonstrated automated compliance checking of spatial geometric relationships applied to building models. Significant opportunities exist to expand the approach more comprehensively across various building domains. Incorporating additional building code systems for analysis will allow the assembly of a more extensive knowledge base.

**Author Contributions:** Methodology, Y.L. and Y.W.; writing—original draft preparation, Y.W. and H.C.; writing—review and editing, X.Z. and Y.W.; supervision, X.Z. and J.W. All authors have read and agreed to the published version of the manuscript.

**Funding:** This work was supported by the Key Research and Development Program of Anhui Province of China (202104a07020014) and the open foundation of the Anhui Province Key Laboratory of Intelligent Building and Building Energy Saving under grant no. IBES2021KF01.

**Institutional Review Board Statement:** Not applicable.

**Informed Consent Statement:** Not applicable.

**Data Availability Statement:** Data sharing is not applicable to this article.

**Conflicts of Interest:** The authors declare no conflict of interest.

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
