# Peer review of "An Automated Fire Code Compliance Checking Jointly Using Building Information Models and Natural Language Processing"

_fire, doi:10.3390/fire6090358_

Round 1

Reviewer 1 Report

There is no doubt that integrating fire egress code compliance checking into BIM is a much-needed innovation in building fire safety. This manuscript introduces a novel approach to automating compliance checks, which could enhance efficiency in building design and safety. However, some concepts and methodologies in the research are built upon existing works.  To enhance the originality of the work, the authors should place more emphasis on the unique algorithms or techniques developed specifically for this system. Therefore, detailed, step-by-step breakdown of these algorithms would provide clarity on how they were developed, their functionality, and what sets them apart.  In addition, a side-by-side comparison that highlights how the developed algorithms differ from or surpass existing ones is needed - the comparison might include aspects like efficiency, accuracy, applicability, and other relevant metrics.

One of the main shortcomings of this manuscript is its lack of a comprehensive review of related works. This limitation hampers its contextualization within the broader fire safety field.  Furthermore, the manuscript lacks a detailed rationale for why these specific algorithms or techniques were developed. To improve, it should explain the problem that these unique algorithms solve and why existing methods were insufficient. 

Another concern is the absence of experimental validation of the developed algorithms. Without concrete evidence, the manuscript will not fully convince readers of their effectiveness. Designing experiments or simulations to test the algorithms against real-world data or scenarios could address this. Please include detailed results, such as performance metrics to  substantiate the originality and effectiveness of the algorithms.

Structurally, the manuscript is organized into logical sections, including introduction, methodology, results, and conclusion. However, some sections are dense and lack clear explanations, posing a challenge for readers unfamiliar with the subject matter. There are also grammatical errors and inconsistencies in terminology.  The manuscript also suffers from a lack of engagement with recent scholarship and a critical analysis of existing literature. The structure of the literature review needs to be improved by organizing it thematically, grouping studies by common themes, methodologies, or findings.

Finally, there are issues related to consistency and citation. Inconsistencies in terminology, formatting, or referencing need to be addressed. Standardizing terminology, following a consistent citation and formatting style, and ensuring uniformity in headings and subheadings would enhance the manuscript's professionalism. The research also needs to clearly state any assumptions and provide justification for them, including empirical evidence or references. 

- The verbiage  is generally grammatically correct but has some inconsistencies in tense and punctuation that could be refined. E.g.  "Automated Compliance Checking (ACC) for construction projects reduces time, cost, and expense errors compared to manual reviews." The phrase  "expense errors" is awkward; it is more appropriate to say "errors in expense calculations.". Some sentences need to be restructured to enhance clarity and readability. e.g. "In the Architecture, Engineering, Construction, and Operation (AECO) industry, construction projects must follow the provisions of the building codes and check their compliance with the requirements of the code." This sentence is quite  long and could be broken down into two sentences for better clarity. The verbiage lacks smooth transitions between ideas, leading to a somewhat disjointed narrative. For example,  the abrupt shift from discussing  innovative integration of BIM to highlighting areas for improvement - there should be transitional sentences to guide the reader through the logic of the argument. Some terms and phrases could be explained for clarity. For example, in lines 86, 97, 90, the authors need to specify what "related works" refers to. There's some repetition of ideas and phrases that could be minimized for a more concise presentation. For example,  "proposed system" is repeated several times; varying the terminology would enhance readability.

- Revise ambiguous or awkward phrasing for clarity.

- Conduct a thorough grammatical review to ensure consistency in tense, punctuation, and terminology.

- Provide more context and break down long awkward sentences.

Author Response

According to the reviewer’s comments, we have revised the manuscript extensively. Pleases see the attachment.

Reviewer 2 Report

This is a very nice paper on automatic compliance checkers. I believe the paper would benefit from improving the clarity of your message - why your approach is better than existing ones, what exact problem it solves and what are the boundaries of its usability. Here are some minor comments:

- the paper requires a rather simple, perhaps graphical explanation of what the "spatial geometries" and "relationships of complex components" are. Your approach resolves issues with them, but I am not sure if I understand these issues correctly. Your solution is mentioned in 3.2, but I think there should be a more general problem description that is clear and somewhere in the introduction.

- p3. I think stating "BIM contains the complete data of the building life cycle" is a bit of an overstatement

- your ACC only verifies the characteristics of building elements and their locations. I think this should be more clearly expressed. From the early problem description, I thought it will be more focused on egress distances etc.

Author Response

(The authors gave the same response as above.)

Reviewer 3 Report

This paper presents An Automated Compliance Checking System for Fire Egress Codes Using BIM, the results seem to be reliable and can be accepted after addressing the following comments:

1.     The literature review needs to be further strengthened, e.g. the state-of-art of BIM

2.     The academic contribution or engineering significance of this paper needs to be highlighted in the abstract, introduction and conclusion.

3.     Chinese characters throughout the text should be translated into English including Figs!

Author Response

(The authors gave the same response as above.)

Reviewer 4 Report

Reviewer Comments:

After reading your manuscript, I am interested in your research on the Automated Compliance Checking (ACC) System for Fire Egress Codes Using Building Information Modeling (BIM). This study proposes an ACC system to resolve the problem of geometric relationships of components in BIM models. In this system, the geometric relationship resolution method guarantees the comprehensiveness of BIM data processing and this system realizes the automatic compliance inspection quickly, accurately and comprehensively. However, I still have some questions and suggestions about this paper as follows:

(1)The specific operational principle of the ACC system in this paper is not sufficiently clear, and how the system achieves its automation needs to be explained within the text.

(2)While the research method in this paper exhibits innovation, the novelty aspect seems insufficient. It is recommended to enhance the exposition of the innovative aspects in the paper.

(3)It is recommended to include a summary or transitional paragraph at the end of the "Related Work" section to better lead into the following content.

(4)The term "SMARTA" mentioned in section 2.2 of the article refers to [please provide the explanation of SMARTA]. It is advisable to provide an annotation for clarification.

(5)It is suggested to label the three sub-tasks involved in the process of constructing the rule checking system in Figure 2, enhancing the visual comprehensibility of the image.

(6)It is recommended to incorporate a comparison of experimental results between the ACC system and other systems in the article, as this can highlight the advantages of this method.

(7)While the paper mentions that the ACC system can achieve rapid, accurate, and comprehensive automated compliance checking, it is limited to analyzing fire-related architectural models. It is suggested to expand the analysis scope to encompass different types of building models to demonstrate the method's applicability.

(8)The visualization in Figure 9 of the article is excessively blurry. It is advisable to substitute it with a clearer image.

(9)Within the ACC system, why is recall rate more important than precision rate?

(10)How many rounds of experiments were conducted in total? Is there any margin for error in the experimental results?

The overall expression of the paper is relatively clear, but there are still some areas that need improvement.The language usage in some sections is a bit convoluted, which could hinder the reader's understanding. For instance, in the 2.2. Industry Foundation Classes (IFC) section,the language usage is quite technical in some sections and could benefit from removing certain specialized terms to make the article more accessible and reader-friendly.

Author Response

(The authors gave the same response as above.)

Round 2

Reviewer 1 Report

The revised manuscript is clearer with improved explanations and scenario validation. However, there are weaknesses with the manuscript -  it does not showcase the method's adaptability across diverse building designs and a comparative analysis with existing methods is still missing. The following improvements must be made:
  1. Elaborate on method's generalizability with concrete examples.
  2. Provide a comparative analysis with existing methods, given that there are numerous studies in the extant literature.
  3. Incorporate "Limitations" section in the manuscript to enable the authors to transparently discuss potential challenges and implications.

English sentences are generally well-constructed, but some have complex structures and should be simplified to aid comprehension. Please use more varied sentence structures to alleviate this. Engage a native English speaker or professional editor to proofread the manuscript to catch any subtle language nuances or minor errors.

Author Response

Thank you again for your careful review. Those comments are valuable and very helpful for revising and improving our paper. We have revised the manuscript accordingly in red font. Please see the attachment.

Reviewer 3 Report

This paper can be accepted

Author Response

Thank you again for your positive comments about our manuscript.

Reviewer 4 Report

Reviewer Comments:

I have reviewed the revised manuscript and I am pleased to confirm that you have diligently addressed the issues in the article as per the suggestions. You have incorporated key term definitions, evaluated existing research achievements and limitations, clarified the operational principles of the ACC system, highlighted the innovative aspects of the study's development, and replaced images for enhanced clarity in presentation.

Moreover, you have iteratively assessed the experiments, minimizing experimental errors and thereby enhancing the persuasiveness of the results. Additionally, language refinement has been undertaken, addressing issues related to wording and expressions in the article.

In conclusion, the modifications and queries I raised earlier have been thoughtfully resolved. The article is now more comprehensive, enriched, and scientifically rigorous. I believe it is suitable for inclusion.

Author Response

(The authors gave the same response as above.)
